# Weighted Blankets’ Effect on the Health of Older People Living in Nursing Homes

**DOI:** 10.3390/geriatrics7040079

**Published:** 2022-07-29

**Authors:** Eva Hjort Telhede, Susann Arvidsson, Staffan Karlsson, Andreas Ivarsson

**Affiliations:** School of Health and Welfare, Halmstad University, 30118 Halmstad, Sweden; susann.arvidsson@hh.se (S.A.); staffan.karlsson@hkr.se (S.K.); andreas.ivarsson@hh.se (A.I.)

**Keywords:** health, nursing home, older people, weighted blanket

## Abstract

Background: An increasingly aging population is a global phenomenon. While considered a positive step forward, vulnerability to age-related health problems increases along with the ageing population. The aim of the study was to investigate weighted blankets’ effect on health regarding quality of life (QoL), sleep, nutrition, cognition, activities of daily living ADL and medication in older people living in nursing homes. Methods: In total, 110 older people were involved in an intervention with weighted blankets, and 68 older people completed the intervention. Measures before and after were performed regarding quality of life; QoL-AD, EQ-VAS, sleep; MISS, nutrition; MNA, cognition; S -MMSE (ADL) and medication. Comparative statistical analyses were applied. Results: After intervention with weighted blankets, health in general, such as QoL, improved. Sleep also improved significantly, especially with respect to waking up during the night. Nutrition was enhanced, health as a cognitive ability improved, and medication in the psychoanaleptic group decreased. The effect size varied between small and large. Conclusions: A weighted blanket seems to be an effective and safe intervention for older people in nursing homes, as several improvements were made regarding the health of older people.

## 1. Introduction

An increasingly aging population is a global phenomenon. While considered a positive step forward, vulnerability to age-related health problems increases along with the ageing population [1]. The meaning of healthy ageing is defined by [2] as “the process of developing and maintaining the functional capacity that enables well-being in old age.” Optimising “functional ability” is this decade’s goal for healthy ageing [2]. Older people’s health and quality of life (QoL) are affected by several factors. An overall fragility increases the risk of mortality, reduces the ability to participate in daily activities and provides limitations on sensory functions and physical strength, which heightens the risk of falls [3,4]. In addition, an overall fragility also results in health-related changes that negatively affect mental health, decrease independence, cause difficulty maintaining personal relationships, decrease sense of meaning and context in life, and reduce opportunities for social support [5,6]. Multiple illnesses, often combined with polypharmacy, also affect the health of older people. These different conditions lead to a greater risk of adverse health effects for older people [7].

The experience of QoL is essential for maintaining the health of older people. Depression, loss of function, physical limitations and pain are associated with reduced QoL. When using a QoL measurement for people with dementia, self and proxy reports can complement each other to address all aspects [8,9,10]. Moving to a nursing home affects older people. They report critical events that negatively affect their health and QoL, for instance, falls, hospital stays, cognitive impairment and loss of loved ones, and this presents a risk of ending up in nursing homes, further reducing QoL [11].

One of the age-related health problems is trouble sleeping [12]. Many of the changes that occur with ageing are related to cognitive functions and abilities where questions have arisen as to whether they are partly a response to sleep deprivation [13]. Older people’s sleep in nursing homes is generally poor [14]. Sleep parameters such as sleep quality, daytime sleepiness and insomnia are associated with cognitive impairment [15]. Short or disturbed sleep and the combination of both also increase the risk of future depressive symptoms and falls in older people [16,17].

Nutritional status is associated with the overall functional ability of older people, and it is a complex aetiology of interacting factors such as sleep problems [12,18]. A good nutritional status is vital to older people’s ability to perform (ADL) [18,19]. Malnutrition is a relevant pathological condition and not entirely uncommon in older people, which causes loss of autonomy, lower QoL, higher frequency of hospitalisations and early death. Older people with impaired nutritional status have risk factors for other serious health problems such as multiple physical disabilities, decreased cognition, and impairment compared to healthy older people living in their own homes [19,20]. Older people in hospitals or nursing homes have an increased risk of malnutrition (50.2%) and measured malnutrition (46.4%) compared to more independent older people [5,6]. Malnutrition is a multifactorial condition often caused by comorbidity, which affects activity functions [21]. Malnutrition leads to reduced health, is associated with poor physical function and can cause an increased risk of falls, anemia, impaired immune system and impaired cognitive status, as well as a greater need for health care [18,22].

When the cognitive ability of older people declines, regardless of the cause, health is also negatively affected. Older people with cognitive impairment have a higher risk of various adverse outcomes such as higher mortality, higher incidence of delirium and dehydration, decreased nutritional status, pain, and impaired physical and cognitive function than more independent older people [5,6,23]. The incidence of behavioural and psychological symptoms of cognitive impairment increases when older people are cared for in hospitals or nursing homes, manifested as increased stress or agitated behaviour. They stay up to six times longer in hospital than other older people and have a greater risk of dying in connection with medical care than more healthy, independent older people [23,24]. Alternative health improvements have tried to affect the cognitive ability of older people using non-pharmacological light therapy, which showed significant beneficial effects on the cognitive decline [25,26].

Even normal ageing can mean a reduction in the functional status of older people. The decline can place the older adult in a negative spiral that leads to additional health problems, affecting their ability to perform (ADL). The ability to perform ADL refers to fundamental activities that are focused on taking care of one’s own body to enable basic needs for health and well-being, such as the ability to act independently regarding baths, toilets, clothing and food. The older people’s health experience is closely related to their ability to perform ADL [27], which plays an essential role in older people’s ability to feel autonomy and thus age healthily [28].

As health problems arise among older people, they lead to increased use of medication. With the high use of medication in older people, the risk of side effects and drug interactions also increases compared to older people who do not utilise as much medication [29]. The negative impact of medication on the body with increasing age is related to changes in body composition and organ function. These physiological age changes cause further changes in medication pharmacokinetics and pharmacodynamic characteristics, giving the opposite effect on older people [30]. Fragile older people in nursing homes are particularly vulnerable to polypharmacy, and polypharmacy is more common in older people in nursing homes than in older people who live in their own homes [31,32]. There is thus a need for alternative medical treatments given the prevalence of multimorbidity, which affects the health of older people, especially when considering that multimorbidity can result in polypharmacy among the older population and consequently increased side effects [33].

One of the alternative non-medical treatments available are weighted blankets. The effect of the weighted blanket is based on theories of deep pressure therapy that originated in the theory of sensory integration. Sensory integration describes how sensory stimuli affect how the brain processes sensory information and can reduce anxiety and stress [10,34]. The effect of deep pressure, as with a weighted blanket, is described as calming, providing better sleep, reducing anxiety and generally increasing well-being. However, few studies have been performed on older people among populations with psychiatric and neuropsychiatric diagnoses [35,36]. This requires research on non-pharmacological methods such as weighted blankets to measure the effect on older people in a nursing home in order to gain increased knowledge and understanding of its impact on the health of older people.

### Aim

The aim of the study was to investigate weighted blankets’ effect on health regarding QoL, sleep, nutrition, cognition, ADL and medication in older people living in nursing homes.

## 2. Materials and Methods

### 2.1. Design

A quasi-experimental design was adopted [37]. 

### 2.2. Participant/Sample

The study included 110 older people > 65 years old who lived in nine comparable nursing homes in municipalities in southwest Sweden. The average age of the participants was 87 years (range 67–99); the proportion of women was 77%. Purposeful sampling was done with the older people. The inclusion criteria were based on sleep problems from the Swedish version of ICD-10, G47.0, which means difficulty falling asleep, sleeping at night or getting enough sleep [38]. Older people with severe lung or heart disease and/or palliative care were excluded from the study. The older people were identified and recruited by the leader of the nursing home and nursing staff. The leaders of the nursing homes made a preliminary request to older people who met the criteria or proxy (relative) for the older people regarding participation in the study, who passed the information to the researcher. The number of older people who discontinued participation in the study was (*n* = 42); this was due to death (*n* = 2), hospital stay (*n* = 2), or the weighted blanket feeling uncomfortable, which was expressed verbally and/or with facial expressions and gestures (*n* = 38). There were 68 remaining older people who participated fully. Mean age in the dropout group was 83 years, while the mean age for those who completed the study was 88 years, indicating a significant difference (*p* = 0.002). The distribution of gender in the dropout group was 57% women. Those who completed the study were 76% women and 24% men (*p* = 0.005). The number of days with a weighted blanket before the withdrawal was, on average, 1.5 days, with intervals of 0–6 days. Those who dropped out came from comparable nursing homes. 

### 2.3. Intervention 

The weighted blanket used in the study was filled with chains and weighed between 4 and 8 kg, about 10% of participant’s body weight, as this weight percentage has been shown to have a calming effect [39]. Most of the older people used the 6 kg weighted blanket. The chains in the weighted blankets were sewn in channels, and the fabric was durable and fireproof. Hygiene covers were not used due to the risk of suffocation. The intervention period was conducted over 28 days, which was based on previous studies where the effect of the weighted blanket was shown after 2–4 weeks [40,41]. If the older people were cold, an ordinary blanket was placed over the older people and then the weighted blanket. The weighted blanket was first tested when the older people were in a normal sleeping position, with the soft side against the older person’s body, starting at the feet. The nursing staff were urged to stay and follow their reaction to observe whether older people could remove the weighted blanket independently. The chains in the weighted blanket were not designed to prevent the older people from moving. If there were no side effects, the weighted blanket was raised farther up the body. The nursing staff was instructed not to place the weighted blanket twice over the chest and not to place the blanket too tightly around the older people’s bodies. The nursing staff was also encouraged to remove the weighted blanket if the older people showed signs of discomfort. 

### 2.4. Measurement 

To estimate the residents’ QoL, the Quality of Life-Alzheimer’s Disease Measure (QoL-AD) was used (Logsdon et al., 2002). This instrument measures the QoL, including people with Alzheimer’s disease, but can also be used by people without Alzheimer’s disease. The scale has 13 items covering the domains of physical health, energy, mood, living situation, memory, family, marriage, friends, chores, fun, money, self and life. It is also possible to answer open-ended questions. Each item is scored on a four-point Likert scale, ranging from 1 (poor) to 4 (excellent), with a possible score range from 13 to 52 [42]. QoL-AD has good validity and reliability, with a Cronbach’s alpha of 0.80 [43]. 

In addition to estimating QoL, EQ-VAS was used. EQ-VAS is a part of EQ-5D, a standardised instrument for measuring and describing the health-related quality of life [44]. EQ-5D contains a vertical visual analogue scale (EQ-VAS) where the older people mark on a scale from 0 to 100. EQ-VAS shows the self-rated health state where 0 is the worst possible health and 100 is the best [44]. EQ-VAS shows high correlations with the MOS SF-20 health perceptions scale (r ¼ 0.70 and 0.72). The test–retest reliability of the EQ-VAS proved to be very high; the intra-class correlation for the VAS was 0.87 [45].

The Minimal Insomnia Symptom Scale (MISS) was used to examine sleep. MISS is an instrument for insomnia that consists of three items: difficulty falling asleep at night, the ability to fall asleep again and the experience of feeling rested when waking up. This is in line with criteria from the International Classification of Diseases, ICD-10, which describes the cardinal symptoms of insomnia as difficulty falling asleep or maintaining sleep or insomnia [46]. Each item is scored on a 4-point Likert scale, ranging from 1 (poor) to 4 (excellent), with a possible score ranging from 13 to 52. MISS shows good sensitivity and specificity in sleep problems and relates to a diagnosis of poor sleep quality based on ICD-10. An ROC analysis identified the optimal cut-off score as ≥7 with a sensitivity of 0.93, specificity of 0.84 and the positive/negative predictive values 0.256/0.995. MISS possessed satisfactory reliability and validity, identified with a Cronbach’s alfa of 0.73 [46,47].

Nutritional status was measured with the Mini Nutritional Assessment-Short Form (MNA-SF). MNA-SF is adapted for older people over the age of 65 and is used to screen the risk of malnutrition [48,49]. MNA-SF consists of six parameters concerning reduced food intake during the last 3 months due to impaired appetite, digestive problems, chewing or swallowing problems, weight loss, physical mobility, mental stress or acute illness, neuropsychological problems and BMI. A score of 12 or greater indicates normal nutritional status, whereas a score of 8–11 indicates ‘at risk of malnutrition’ and a score of 7 or less malnutrition. All parameters scored from 0 to 2 or 3 with a total score of 0–14 [48,49]. MNA-SF is a validated instrument with high sensitivity and specificity [48]. Correlation between MNA-SF and full MNA is high (Pearson’s r = 0.969) [50].

A Standardised Mini-Mental State Examination (S-MMSE) instrument consists of 20 questions divided into 11 areas, with a maximum score of 30. The questions cover orientation to time and space, memory, language, and visuospatial functions that relate to visual and spatial interpretive ability, time orientation, and immediate reproduction. A score of >20 indicates an indication of normal cognitive function. Scores of <20 indicate the presence of cognitive impairment [51,52,53]. S-MMSE is used as a clinical screening test for cognitive impairment, with good reliability [51,54]. The internal consistency obtained by Cronbach’s alpha shows 0.826 [55]. 

Katz ADL index is a standardised measure for evaluating treatment, prognosis and functional changes in older people and people with chronic illnesses in institutionalised settings [56]. The instrument describes dependence on personal activities of daily living (PADL) [57] (pp. 171–178). The Katz ADL index is specially developed to measure activity in older people based on six activities in daily life. Each activity is graded with dependence, independence, and partial dependence on activities regarding food intake, continents, movement, toilet visits, dressing and undressing, showering and bathing. A dichotomisation of independent or dependent can be carried out by considering shower/bath, dressing/undressing and food intake as independent and activity’s toilet visits, movement and continence as dependent. Six points indicate full independence and two points or less dependence [58,59]. Katz ADL index shows a high internal consistency with a Cronbach’s alpha of 0.838 [52].

Mapping of the resident’s medication use was carried out based on the Anatomical Therapeutic Chemical Classification ATC-Kode. Adopted by the WHO, this allows pharmacology to be divided into different groups according to the indication area. The drug doses collected were based on the name of the generic pharmacy drug power per mg [60]. 

### 2.5. Procedure 

The data collection took place from 2019 to the summer of 2021 in nursing homes during the daytime. A preliminary request for participation was sent to older people or a proxy together with written information about the aim of the study and the consequences of being included. Nursing home leaders informed the researcher which older people had agreed the researchers could contact them to inform them about the study and acquire their consent to participate. The researcher then sent informed consent forms to the older people or their proxy to acquire a signature. In connection with the start of the study, oral information was given again, and oral consent was obtained. A baseline assessment of the older people was conducted using the instruments. The questions in the instruments were assessed by the older people themselves and the researcher in cases where the older people had the cognitive ability to answer independently. In cases where older people had an S-MMSE rating of four or lower, it was assessed on the basis of what the older people would do according to the nursing staff who knew the older people as well as a representative (a person-proxy perspective), together with the researcher. If older people had S-MMSE between 4 and 10, collection for the instrument was performed with the proxy and the older people. After 28 days, the weighted blanket was removed. The same day the weighted blanket was removed, data were collected with the instrument for a post-measurement to assess the outcome of the weighted blanket. 

### 2.6. Analysis 

To determine the sample size required to obtain adequate power, a priori sample size calculation was performed using Power Calculation [61]. In this calculation, the specified alpha was 0.05 and Beta 0.80, resulting in a required sample size of 34 older people. Due to the high risk of dropout within this population, a total of 110 persons were enrolled in the study, and 68 older people completed the study. Descriptive statistics [62] were used to describe the study group. Comparative statistical analyses were used to examine health differences at the individual level and between groups regarding QoL, sleep, nutrition, cognitive ability, ADL ability and medication concerning the intervention with a weighted blanket. The medication use was summarised 28 days before the intervention and 28 days during the intervention on the last day of weighted blanket use. The variables of the QoL scale were divided into four domains: behavioural competence meant physical health; psychological well-being was about mental health; the third domain, perceived QoL, involved evaluating family and friends; and the environmental domain included housing quality and the ability to perform housing duties [63]. Due to the lack of a control group, comparisons were made between measurements before and after the intervention, and the older people were their own control group. To compare the mean value of customarily distributed variables from baseline to post-measurement, a paired-sample student *t*-test was used, where the sample was normally distributed. For all analyses, a *p*-value < 0.05 was considered statistically significant [62]. Cohen’s d was used as an effect size to indicate the magnitude of the effects for the comparative analyses. To interpret the effect sizes, the suggested cut-offs were small = 0.2, medium = 0.5 and large > 0.8 [62] (pp. 114–116).

## 3. Results

Baseline measurement on the summary of older people’s QoL was 26.4 points, with a possible total range from 13 to 52 [42]. Compared to baseline, QoL as summary improved after using a weighted blanket for 28 days (*p* < 0.001, d = 0.78). The results also showed a statistically significant increase in behavioural competence (*p* = 0.003, d = 0.37) and environmental quality (*p* < 0.001, d = 0.76). In addition, self-rated health-related quality of life increased significantly (*p* < 0.001, d = 0.61); see Table 1.

The limit value for sleep problems in older people is considered to be ≥7 points [48], the baseline measurement for the group of older people was 6.9 points. In summary, sleep quality increased (*p* < 0.001, d = 0.68), waking up (*p* < 0.001, d = 1.10) and sleep latency (*p* < 0.001, d = 0.43) all improved after using a weighted blanket for 28 days compared to baseline. The group at baseline measurement showed 7.1 points, which indicated a risk of malnutrition. The nutritional status improved significantly in summary (*p* < 0.001, d = 0.44). The results also showed a statistically significant increase in food intake (*p* < 0.001, d = 0.42); see Table 2. 

Baseline measurement regarding the cognitive ability of the older people was 8.6 points, which indicated severe cognitive impairment, as severe dementia is considered between 0 and 9 points [51]. In the group, 22 (32%) older people had moderate dementia, and 39 (57%) severe dementia. The cognitive ability improved in summary (*p* < 0.001, d = 0.51), and regarding orientation (*p* = 0.002, d = 0.46), it increased significantly after weighted blanket use for 28 days compared to baseline; see Table 3. 

Dependency in ADL was most common regarding continence, toileting and transfer, but there were no significant differences between baseline and after weighted blanket. Compared with the baseline, medication use in the psychoanaleptic group decreased significantly (*p* = 0.014, d = 0.29); see Table 4 and Table 5.

## 4. Discussion

The use of weighted blankets provides various improvements in older people’s QoL. In this study, the weighted blanket improved the health in QoL regarding behaviour, anticipation, and psychological and mainly environmental aspects. The domain environment includes increased participation in tasks and daily activities at the nursing home, primarily concerning housing activities. Older people who experience a high QoL have shown participation and activity to be crucial in maintaining health [9,10]. There are limited previous studies on the age group of older people who have examined the effect of weight blankets on QoL. The previous studies have mainly studied children with ADHD and/or ASD and participants with psychiatric disorders [61,62,63,64]. However, in those groups, the weighted blanket showed an improvement in the activities around the morning and evening routines that facilitated the ability to master everyday life, which led to a higher level of well-being and health [64,65,66]. The term QoL is comprehensive and describes several parameters in life, where physical ability is included as a prerequisite for a possible high QoL [67]. In addition, low QoL is associated with a higher mortality rate, even among initially healthy elderly [68]. There is an advantage in improving QoL in older people. The environment around older people can create conditions for better health where changes are made by creating interventions that can provide conditions for a better quality of life [69]. It is possible to predict that interventions with weighted blankets can increase the QoL in many domains for older people living in nursing homes. More studies on this age group are needed to clarify the relationship between the weighted blanket and improved QoL.

Weighted blankets can be a valuable aid in improving sleep. The results of this study showed that the number of awakenings during the night decreased, and less awakening resulted in a more cohesive night’s sleep. It also became more manageable for older people to fall asleep in the evening. Previous studies have also described this overall improved effect on sleep using the weighted blanket [40,70,71] Insomnia in older people is associated with cognitive impairment [15]. A weighted blanket can also affect depressive symptoms, which are alleviated in connection with sleep problems and vice versa [17]. The weighted blanket is considered to dampen the nervous system’s stress system via its deep pressure, which may be the reason for the calming effect that improves sleep for older people [35,72,73]. However, some studies do not support this finding regarding improved sleep. Regardless, they describe the anti-anxiety effects of weighted blankets [74]. The deep pressure that the weighted blanket causes create effects and anxiety suppression that are considered to predict improved sleep. The heart rate is also lowered, which creates the conditions for improved sleep [75]. The weighted blankets’ prerequisites to improve sleep are essential in clinical practice in nursing homes, where sleep problems are common among older people. Despite the high proportion of discontinuations in the study, the beneficial effects of the weighted blanket still outweigh the discomfort that the weighted blanket caused some older people. Those older people who experienced discomfort with the weighted blanket showed it clearly and directly in the intervention, which reduced the risk of someone using the weighted blanket against their will. The weighted blanket seems to be a relatively simple way, with minimal risks, to help older people with sleep problems. 

A weighted blanket may decrease the utilisation of inappropriate medication with a high level of side effects for older people. The current study showed that mainly psychohanaleptics decreased with the weighted blanket. The reduction in psychoanaleptics is significant given that psychoanaleptics are not recommended for older people due to an increased risk of serious side effects [76,77,78,79,80]. Alternative interventions that reduce the use of medication in older people are essential, given that the combination of multimorbidity and polypharmacy is common and leads to a greater risk of adverse health effects [7,31]. One study even highlighted the weighted blanket as a possible alternative to medication [75], and it has been shown that nursing staff stated that older people’s medication use decreased in connection with using weighted blankets [71]. There is a link between increased health problems in older people and increased medication use. With the high use of medication in older people, side effects and medication interactions also increase [29]. Weighted blankets could contribute to being a non-pharmacologically safe and clinically meaningful alternative for older people in nursing homes. However, more studies in this age group need to be conducted to understand the effect of weighted blankets on older people. 

### Methodological Considerations 

There is a weakness in the internal validity given that the older people who had cognitive decline did not respond independently and had to rely on proxy assessment. However, the strength increased, considering that the proxy knew the older people well. There was also a risk that used instruments were misinterpreted by cognitively healthy older people and proxies. However, the instruments used were validated and previously tested on older people, strengthening the reliability. Conducting studies in which people with cognitive weakness participate is a strength, as many older people in nursing homes have a cognitive weakness. The researcher was involved in filling in the instruments together with older people during the study period despite remaining neutral. In cases where the questions in the instruments were clarified, the researcher left time for older people to answer the questions without interference or disturbance.

There is a weakness related to how well the nursing staff followed the instructions for using and handling the weighted blanket. Oral and written guidelines were given about how the weighted blanket should be implemented, which was a strength. The researcher also made regular visits to nursing homes to follow up on the implementation of the intervention. In connection with the COVID-19 pandemic, it was impossible to carry out on-site visits to the same extent as before, which was a weakness. However, the researcher conducted a telephone follow-up instead, which added strength. 

The study included nine separate nursing homes in different places in southwest Sweden. Nevertheless, a weakness for objectivity is that the study only included 68 older people. The power analysis indicated a sample of 34 older people for significant results, but the larger sample in the study provided more strength in conclusion, which also reinforced the objectivity [37].

The proportion of older people who dropped out of the study may have affected the results. The proportion of men was higher, and the average age was lower in the dropout group than in the participant group. The study was conducted with a comparative analysis, baseline measurement and post-measurement, which was considered useful as older people were their own control group; in that way, differences at the group level were examined [62] (pp. 521–570).

## 5. Conclusions

A weighted blanket seems to be an effective and safe intervention for older people in nursing homes, as several improvements were made regarding the health of older people, especially regarding the improved quality of life and sleep. The weighted blanket could be a beneficial non-pharmacological intervention for older people, as this study showed a reduction in the utilisation of medication for the group, significantly so for psychoanaleptics. Thus, the study results can, in clinical work, help older people maintain health. Furthermore, the intervention with a weighted blanket is safe to use and influences many parts of older people’s lives. Therefore, the weighted blanket can be an intervention to improve the health of older people. This study showed that the use of weighted blankets had an effect on health regarding the quality of life, sleep, nutrition, cognition, ADL and medication in older people in nursing homes. To further clarify the impact of the weighted blanket on health, more and larger studies are needed to clarify the effects of the weighted blanket on older people in a longitudinal view.

## Figures and Tables

**Table 1 geriatrics-07-00079-t001:** QoL before and after utilisation of weighted blanket for 28 days, *n* = 68.

QOL-AD, Mean (sd)	Before Weighted Blanket	After Weighted Blanket	*p*-Value ^1^	Effect Size ^2^
Behaviour	8.6 (2.3)	9.5 (1.9)	0.003	0.37
Environment	6.3 (1.4)	7.5 (1.2)	<0.001	0.76
Anticipation	9.1 (10.7)	8.4 (1.9)	0.586	0.35
Psychological	3.9 (0.9)	4.2 (0.9)	0.043	0.25
Sum	26.4 (4.5)	30.1 (3.9)	<0.001	0.78
EQ VAS, Mean (sd)	52.3 (15.4)	59.4 (12.8)	<0.001	0.61

^1^ Student’s *t*-test; ^2^ Cohen’s.

**Table 2 geriatrics-07-00079-t002:** MNA before and after utilisation of weighted blanket for 28 days, *n* = 68.

	Before Weighted Blanket	After Weighted Blanket	*p*-Value ^1^	Effect Size ^2^
MISS, Mean (sd)				
Sleep latency	2.2 (1.2)	1.7 (0.9)	<0.001	0.43
Waking up	2.8 (1.0)	1.6 (0.8)	<0.001	1.10
Well rested	1.9 (1.0)	1.9 (0.8)	0.909	0.01
Sum	6.9 (2.8)	5.4 (2.2)	<0.001	0.68
MNA, Mean (sd)				
Reduced food intake	1.5 (0.7)	1.8 (0.5)	<0.001	0.42
Weight loss	1.9 (0.9)	2.0 (1.0)	0.387	0.11
Mobility	0.2 (0.8)	1.1 (0.8)	0.106	0.20
Mental stress	0.3 (0.4)	0.6 (0.5)	<0.001	0.47
Neuropsychological problem	0.6 (0.6)	0.6 (0.6)	0.698	0.05
BMI	1.8 (0.6)	1.9 (0.6)	0.382	0.26
Sum	7.1 (2.0)	8.1 (1.6)	<0.001	0.44

^1^ Student’s *t*-test; ^2^ Cohen’s d.

**Table 3 geriatrics-07-00079-t003:** S-MMSE before and after utilisation of weighted blanket for 28 days, *n* = 68.

	Before Weighted Blanket	After Weighted Blanket	*p*-Value ^1^	Effect Size ^2^
S-MMSE, Mean (sd)				
Orientation	1.0 (1.5)	1.4 (1.6)	0.001	0.46
Registration	1.6 (1.3)	1.9 (1.2)	0.002	0.38
Attention/calculation	2.3 (2.1)	2.8 (2.0)	0.006	0.34
Recall	0.6 (1.0)	0.9 (1.1)	0.002	0.38
Langue	3.0 (2.3)	3.4 (2.4)	0.146	0.30
Sum	8.6 (7.4)	10 (7.5)	<0.001	0.51

^1^ Student’s *t*-test; ^2^ Cohen’s d.

**Table 4 geriatrics-07-00079-t004:** Percentage movement in ADL dependent before and after utilisation of weighted blanket for 28 days, *n* = 68.

	Before Weighted Blanket	After Weighted Blanket	*p*-Value ^1^
Dependent, %			
Bathing	47	43	0.375
Dressing	48	44	0.250
Toileting	72	72	1.000
Transfer	60	56	0.453
Continence	75	69	0.219
Eating	12	12	1.000

^1^ Wilcoxon signed rank test.

**Table 5 geriatrics-07-00079-t005:** Medication, before and after utilisation of weighted blanket for 28 days, *n* = 68.

	Before Weighted Blanket	After Weighted Blanket	*p*-Value ^1^	Effect Size ^2^
Medication, Mg, Mean (sd)				
Analgetica	31,600 (38,597)	31,268 (37,955)	0.849	0.01
Neuroleptica	967 (4338)	445 (1308)	0.256	0.14
Hypnotics	43 (82)	42 (81)	0.321	0.12
Psykoanaleptica	975 (1275)	938 (1273)	0.014	0.29

^1^ Student’s *t*-test; ^2^ Cohen’s d.

## Data Availability

Data are available on request due to restrictions, e.g., privacy or ethical. The data presented in this study are available on request from the corresponding author. The data are not publicly available due to sensitive personal data regarding the health of study participants.

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
