# Peer review of "Weighted Blankets’ Effect on the Health of Older People Living in Nursing Homes"

_geriatrics, 2022, doi:10.3390/geriatrics7040079_

Round 1
Reviewer 1 Report
The manuscript presents an interesting topic.
However, there are issues that need to be addressed:
- It is not clear why the authors included nutritional status assessment in the present study. Please explain the association between nutritional status with the use of weighted blankets.
- In the discussion, the high proportion of discontinuation of the intervention should be clearly addressed. Would the discomfort caused by the blanket outweigh its benefits for older adults?
- The conclusion should contain an objective response to the aim of the study.
Author Response
|
Reviewers 1 comments
|
Authors reply |
|
1. The manuscript presents an interesting topic
|
Thank you very much for your feedback. |
|
2. It is not clear why the authors included nutritional status assessment in the present study. Please explain the association between nutritional status with the use of weighted blankets. |
Thank you very much for your comment. Changes have been made so that it is now more apparent how the connection between nutrition and sleep can be linked. Line 60-61
|
|
3. In the discussion, the high proportion of discontinuation of the intervention should be clearly addressed. Would the discomfort caused by the blanket outweigh its benefits for older adults? |
Thank you very much for your comments. Changes have been made based on your comment. Line 319-323 |
|
4. The conclusion should contain an objective response to the aim of the study. |
Thank you very much for your comment, we have clarified in the conclusion text. Line 369-371 |
Reviewer 2 Report
The manuscript presents a clear study design and findings. Minor changes that may improve readability include following:
- please review referencing styles, such as reporting page numbers for direct quotes (e.g., line 33-34)
- an additional round of editing would be beneficial - for example, line 65 refers to "increased" level of malnutrition, however, comparison base is not clear
- line 83 seems to use ADL as an abbreviation first time, which usually requires spelling it out
- line 168 uses a combination of [ and (, which is an unusual case
- line 176 refers to "insured" appetite, which is not a common combination of words
- general review for the purposes of consistency - lines 264 and 265 present two different ways to report a p-value
- finally, the authors may consider merging tables, but this is not critical
Overall, with exception of very minor English language changes, the paper reports the problem, the methodology, results and conclusions very clearly.
Author Response
|
Reviewers 2 comments
|
Authors reply |
|
1. The manuscript presents a clear study design and findings |
Thank you very much for your feedback |
|
2. Please review referencing styles, such as reporting page numbers for direct quotes (e.g., line 33-34 |
Thanks for your comment. Changes have been made established on your comment, when applicable. |
|
3. an additional round of editing would be beneficial - for example, line 65 refers to "increased" level of malnutrition, however, comparison base is not clear |
Changes have been made established on your comments when applicable
|
|
4. Line 83 seems to use ADL as an abbreviation first time, which usually requires spelling it out |
Thank you very much for noticing that. It has been changed to “activities of daily living (ADL)” on line 61. |
|
5. Line 168 uses a combination of [ and (, which is an unusual case |
Thank you very much for noticing that. We changed it when double marking was used. We use [ for references and (for another numbering
|
|
6. Line 176 refers to "insured" appetite, which is not a common combination of words |
Thank you very much for noticing that “insured” has been changed to “impaired”. |
|
7. General review for the purposes of consistency - lines 264 and 265 present two different ways to report a p-value |
Thank you very much for noticing that it has been changed |
|
8. Finally, the authors may consider merging tables, but this is not critical |
Thank you for the comment. We divided the tables after each presentation in text, as we considered it helped the reader to follow the results. |
|
9. Overall, with exception of very minor English language changes, the paper reports the problem, the methodology, results and conclusions very clearly. |
Many thanks for your feedback. |
Round 2
Reviewer 1 Report
The authors addressed all the issues indicated in the first review.
Therefore, there are no further comments.